# Effect of Vitamin D_3_ Supplementation vs. Dietary–Hygienic Measures on SARS-CoV-2 Infection Rates in Hospital Workers with 25-Hydroxyvitamin D3 [25(OH)D3] Levels ≥20 ng/mL

**DOI:** 10.3390/microorganisms11020282

**Published:** 2023-01-21

**Authors:** Maria Elena Romero-Ibarguengoitia, Dalia Gutiérrez-González, Carlos Cantú-López, Miguel Ángel Sanz-Sánchez, Arnulfo González-Cantú

**Affiliations:** 1Department of Research, Hospital Clínica Nova, San Nicolás de los Garza 66450, NL, Mexico; 2Departamento de Ciencias Clínicas, Vicerrectoría de Ciencias de la Salud, Universidad de Monterrey, San Pedro Garza García 66238, NL, Mexico; 3General Management, Hospital Clínica Nova, San Nicolás de los Garza 66450, NL, Mexico; 4Department of Endocrinology, Hospital Clínica Nova, Av. del Bosque 139, Cuauhtémoc, San Nicolás de los Garza 66450, NL, Mexico

**Keywords:** SARS-CoV-2, vitamin D3, supplementation

## Abstract

**Background**: There is scant information on the effect of supplementation with vitamin D3 in SARS-CoV-2 infection cases when patient 25-hydroxyvitamin D3 [25(OH)D3] levels are between 20–100 ng/mL. We aimed to evaluate the effect of supplementation with vitamin D3 vs. dietary–hygienic measures on the SARS-CoV-2 infection rate in participants with serum 25(OH)D3 levels ≥20 ng/mL. **Methods**: This study was quasi-experimental. We invited hospital workers with 25(OH)D3 levels between 20–100 ng/mL and no previous SARS-CoV-2 infection. They were randomized as follows: treatment options were a) vitamin D3 supplementation (52,000 IU monthly, G1) or b) dietary–hygienic measures (G2). We conducted a 3- to 6-month follow-up of SARS-CoV-2 infections. Participants with 25(OH)D3 levels <20 ng/mL were also analyzed. We divided these latter participants depending on whether they were supplemented (G3) or not (G4). **Results:** We analyzed 198 participants, with an average age of 44.4 (SD 9) years, and 130 (65.7%) were women. G1 had fewer cases of SARS-CoV-2 infection than G2 after a follow-up of 3- to 6-months (*p* < 0.05). There were no differences between G3 and G4 at the 3- and 6-month follow-up cutoff points (*p* > 0.05). Using a mixed effect Cox regression analysis in the 164 participants that completed six months of follow-up, vitamin D3 supplementation appeared to act as a protective factor against SARS-CoV-2 infection (HR 0.21, *p* = 0.008) in G1 and G2. None of the participants treated with the supplementation doses had serum 25(OH)D3 levels >100 ng/mL. **Conclusions:** Vitamin D3 supplementation in participants with 25(OH)D3 levels between 20–100 ng/mL have a lower rate of SARS-CoV-2 infection compared with the use of dietary–hygienic measures at six months follow-up.

## 1. Introduction

COVID-19 is an infectious disease caused by the newly discovered coronavirus SARS-CoV-2; its clinical spectrum ranges from asymptomatic infection to critical and fatal illness [1,2]. The first case was reported in Wuhan, China, in December 2019, while the first case in Latin America was detected in Brazil in February 2020 [3,4]. As of 13 March 2022, approximately 458,479,635 cases had been confirmed, and 3 million deaths were reported worldwide. Furthermore, 10,712,423,741 vaccine doses have been applied [5].

To date, prevention remains the cornerstone of management to decrease infection rates. In late 2020, the authorization of SARS-CoV-2 emergency vaccines led to partial pandemic control. However, further studies will be key to obtain clear evidence on its treatment and prevention [6].

There is controversy on the use of vitamin D3 supplementation in the prevention of SARS-CoV-2 [7,8]. A meta-analysis concluded that a low serum 25-hydroxyvitamin D3 [25(OH)D3] level was significantly associated with a higher risk of SARS-CoV-2 infection [9]. However, there is limited data on the link between SARS-CoV-2 infection and vitamin D3 supplementation in individuals with normal 25(OH)D3 levels.

The rationale for this study was to establish a relationship between vitamin D_3_ supplementation and the incidence of SARS-CoV-2 infection in a prospective study. We hypothesized a positive effect of vitamin D supplementation on the immune system (innate and adaptive immunity) [10,11,12], and that the group who underwent supplementation would develop fewer SARS-CoV-2 cases.

The study aimed to evaluate the effect of vitamin D3 supplementation vs. dietary–hygienic measures on the development of SARS-CoV-2 infection in participants with serum 25(OH)D3 ≥20 ng/mL (primary outcome). Secondarily, we compared a group of hospital workers with serum 25(OH)D3 <20 ng/mL that received or did not receive supplementation with vitamin D3.

## 2. Methods and Materials

The study was a prospective, quasi-experimental study that followed the CONSORT guidelines [13]. The target population included health workers at high risk of SARS-CoV-2 infection and vitamin D serum values ≥20 ng/mL from a hospital in Northern Mexico, the Hospital Clínica Nova (HCN), at a northern latitude of 25°45′ and western latitude of 100°17′. This hospital was converted into a COVID-19 hospital in March 2020. The participant recruitment began in May through to August 2020, and the follow-up continued from August 2020 through to January 2021.

The study was conducted per the Code of Ethics of the World Medical Association (Declaration of Helsinki) for experiments in humans. Due to the study’s nature, each participant and two witnesses signed an informed consent form. The author, AGC, enrolled the participants and assigned them to each intervention.

The inclusion criteria were age between 18 and 65 years old, both genders, absence of infection by SARS-CoV-2 at the time of serum vitamin D3 determination, lack of infection at any site (bacteria or fungi), and participants had to be hospital workers. In addition, participants were excluded if their serum 25(OH)D3 was >100 ng/mL, if they had previously received supplements containing vitamin D3, and if they were pregnant.

After signing the consent form, participants were directed to the laboratory to provide serum samples to determine their levels of 25(OH)D3. We used the Elecsys total vitamin D test with COBAS 6000 (e601) equipment (Wiesbaden, Germany). The variation coefficient was 4.1%; the analytic specificity for 25(OH)D3 was 100%, and the analytic sensibility was established at 4.01 ng/mL. The laboratory staff performed calibration of the equipment every time there was a change in the lot, and the calibration curve factor was 1 (goal 0.8–1.2).

Based on the results, we classified participants into four groups. The first and second groups had 25(OH)D3 >20 ng/mL, and they were randomized in a 1:1 ratio. The participants had two treatment options: the first was supplementation with vitamin D3 52,000 IU in a single dose, monthly, for three months (13 tablets of 4000 UI, G1). The total dose was based on the Endocrine Society’s Guidelines [14,15]; however, we decided to administer it every month instead of a daily, since some of this frequency’s benefits have been discussed elsewhere [16]. The second option was treatment based on dietary–hygienic measures (G2), such as sun exposure for at least 10 min per day between 10:00–18:00 h, and foods that were rich in vitamin D3 and D2 (fish, meat, eggs, milk, mushrooms, and almonds) [17,18].

The third and fourth groups had 25(OH)D3 levels <20 ng/mL. These groups were not part of the randomization process, since we considered that all subjects with vitamin D3 <20 ng/mL should receive vitamin D3 supplementation [19], and they were referred to their personal primary care physician (different primary care physician) for treatment and follow-up (G3). Nevertheless, a certain number of participants decided, of their own volition, not to receive vitamin D3 supplementation (G4). Since they were health workers and had medical records from the same hospital where the study was conducted, we could still follow them over time. We, therefore, had four groups for comparison.

Variables obtained from the medical record were age, gender, occupation, diabetes mellitus, hypertension, allergies, asthma, smoking history, previous hospitalizations, and BMI.

The participants were monitored monthly during follow-up, by telephone, and every three months in a face-to-face interview. We inquired about COVID-19 symptoms (myalgias, hyposmia, cough, malaise and fatigue, nasal congestion, fever, diarrhea, thoracic pain, shaking chills, nausea, and vomiting) and whether they had been diagnosed with SARS-CoV-2 infection by serologic or swab tests (PCR). Furthermore, every month patients were asked if they had been consuming food with vitamin D3 or D2 using a 24 h food recall questionary. Serum 25(OH)D3 was measured at baseline and after three months of follow-up.

The relative risks that could be present in the different groups were increased serum vitamin D levels (>100 ng /mL) in G1 and decreased vitamin D serum levels <20 ng/mL in G2. The research group could not control the risks in G3 and G4.

## 3. Statistical Analysis

Two researchers reviewed the quality control of the database and anonymized it. The normality assumption was evaluated with the Shapiro–Wilk test and frequency histograms. Descriptive statistics were computed, such as the mean, the standard deviation for quantitative variables, frequencies, and percentages for categorical variables. Chi-square tests and ANOVA were used to compare the categorical and quantitative data between groups. Kaplan–Meier curves and the log-rank test were used to evaluate the difference in SARS-CoV-2 infection between groups. We conducted a mixed-effect Cox regression analysis in the groups with 25(OH)D3 ≥20 ng/mL and <20 ng/mL, in which the dependent variable was SARS-CoV-2 infection at six months. The covariates computed in the model were vitamin D3 supplementation, age, and gender. The seasonal variation was a random effect. Missing data were handled by complete case analysis. For simple randomization, we used random number generation with a binomial distribution and a probability of 50%. The author, AGC, generated the randomization sequence.

The statistical programs used were SPSS version 25 (IBM, Armonk, NY, USA) and R software version 4.0.3 (R Core Team, Vienna, Austria). The analysis was two-tailed. A *p*-value <0.05 was considered statistically significant. The sample size included all hospital workers that agreed to provide serum samples for 25(OH)D3 testing.

## 4. Results

Initially, 205 hospital workers were considered for the study; five had to be excluded because they had previously contracted SARS-CoV-2 before the study started. Additionally, two individuals in G1 and G2 withdrew their consent to participate.

In the end, 198 participants were analyzed. These participants were distributed into four groups based on their 25(OH)D3 levels and the type of treatment administered. Participants in G1 received a supplementation of 52,000 IU/month for three months (n = 43). Individuals in G2 had dietary–hygienic measures (n = 42). G3 received an average dose of 90,000 IU/month for three months, provided by their treating physician (n = 28). Finally, G4 did not receive supplementation (n = 85).

The mean (SD) age in the four groups was 44.4 (9.1) years (*p* > 0.05), with no difference between groups, and 130 were female (65.7%). The three most frequent professions in all the groups were: physicians, nurses, and administrative workers, and no difference was established among G1–G4 (*p* > 0.05). There were also no differences in terms of smoking history, body mass index (BMI), allergies, Type 2 diabetes, hypertension, and asthma (*p* > 0.05). Demographic data and medical history of the groups are described in Table 1.

The mean (SD) serum baseline 25(OH)D3 levels reported per group were: 27.1 (6.7) ng/mL in G1, 26.6 (5.5) ng/mL in G2, 15.4 (3.4) ng/mL in G3, and 14.9 (3.1) ng/mL in G4, *p* < 0.001. Serum levels of 25(OH)D3 were again measured after three months of follow-up. The values were as follows: G1 33.8 (7.1) ng/mL, G2 22.4 (6.9) ng/mL, G3 38.2 (8.5) ng/mL, and G4 22.1 (5.8) ng/mL, *p* < 0.001. None of the participants had 25(OH)D3 levels >100 ng/mL and no adverse events were reported.

At the 3-month cutoff, 51 (25.8%) of the 198 workers had developed SARS-CoV-2 infection (naïve variant). The distribution of cases by groups was: G2 14 (33%) and G1 3 (7%), *p* = 0.002; G4 27 (32%) and G3 7 (25%), *p* > 0.05. When comparing between the four groups there was a statistical difference, *p* = 0.017.

One hundred and eighty-seven (187) individuals were followed for four months; of these, 56 (29.4%) developed a SARS-CoV-2 infection (naïve variant). The proportion of SARS-CoV-2 infection cases by groups was: G2 14 (34%) and G1 6 (14%), *p* = 0.002; G4 29 (38%), and G3 7 (26%), *p* = 0.04. When comparing between the four groups there was a statistical difference, *p* = 0.041

Additionally, we studied 167 workers that completed a 6-month follow-up. Not all participants completed this follow-up because they were recruited later and received SARS-CoV-2 vaccination, and we considered that it could affect our results. However, during that period, 56 (33.5%) developed COVID-19 (naïve variant). The number of cases by the groups were as follows: G2 13 (39%) and G1 5 (13%), *p* < 0.001; G4 29 (40%) and G3 9 (37%), *p* >0.05. When comparing the four groups, there was a statistical difference, *p* = 0.031. Figure 1 shows comparisons between G1 vs. G2 and G3 vs. G4. Table 2 shows SARS-CoV-2 infection by groups, symptoms, and outcomes at six-month follow-up. There were no deaths in this study.

Kaplan–Meier curves of the SARS-CoV-2 infection rate during the six-month follow-up. The images compare the four groups according to their baseline vitamin status and vitamin D3. Figure 1A shows the group with serum 25(OH)D3 >20 ng/mL where there is statistical difference in SARS-CoV-2 infection rate. Figure 1B shows 25(OH)D3 <20 ng/mL where there was no statistical difference in infection rate. The lowest rate of SARS-CoV-2 infection occurred in the group with vitamin D >20 ng/mL plus supplementation for three months, at 52,000 IU per month.

We conducted a mixed-effect Cox regression analysis with the participants who were followed for six months (n = 164). The final covariates computed in the model were vitamin D3 supplementation, age, and gender. The seasonal variation was a random effect. In the Group ≥20 ng/mL 25(OH)D3, the resulting HR was 0.21 (SE 0.58, *p* = 0.008) for vitamin D3 supplementation; age, 0.97 (SE 0.26, *p* = 0.25); and gender, 0.85 (SE 0.49, *p* = 0.76). In the Group <20 ng/mL 25(OH)D3, the HR was 1.15 (SE 00.39, *p* = 0.72) for vitamin D3 supplementation; age, 1.001 (SE 0.02, *p* = 0.94); and gender, 0.52 (SE 0.48, *p* = 0.18). Other covariates that were explored were BMI, Type 2 diabetes, and hypertension, but they were not statistically significant, so they were eliminated from the final models.

## 5. Discussion

This study demonstrated that vitamin D3 supplementation for three months led to a decrease in the rate of SARS-CoV-2 infection in the group of participants with 25(OH)D3 levels ≥20 ng/mL throughout the 3–6-month follow-up when compared with dietary–hygienic measures.

In this study, participants with 25(OH)D3 levels <20 ng/mL had a higher rate of SARS-CoV-2 infection; however, upon the comparison of groups (G3 and G4), the supplemented group had less frequent SARS-CoV-2 infections at four months of follow-up but not at six months.

Previous studies revealed an association of low serum 25(OH)D3 levels with SARS-CoV-2 infection, aging, and comorbidities such as diabetes mellitus, hypertension, or obesity [9,20,21]. We did find an association between low levels of 25(OH)D3 and SARS-CoV-2 infection; however, there was no association with comorbidities.

A study conducted in Barcelona demonstrated that participants that had previously received vitamin D for four months were at a decreased risk of acquiring SARS-CoV-2 infection (HR = 0.95, CI 0.91–0.98) [22]. A previous clinical trial in Mexico City included health workers randomized to receive either 4000 IU of vitamin D3 for 30 days or a placebo and were followed for 45 days. The results showed that independent of the baseline 25(OH)D3 (that in the study was mostly deficient), the supplemented group had a lower incidence of SARS-CoV-2 infection [23].

Other studies have shown an inverse dose-response relationship between continuously increasing 25(OH)D concentrations (from 15 to 60 ng/mL with supplementation) and a parallel decreased probability of requiring COVID-19-related hospitalization; also, a trend in decreasing mortality, ICU admission, invasive and non-invasive ventilation [8,24].

Our study supports the importance of supplementation for the prevention of SARS-CoV-2 infection. Nevertheless, there is still a lack of information on the potential benefit of supplementation with vitamin D3 in participants with 25(OH)D3 ≥ 20 ng/mL to decrease SARS-CoV-2 infection risk; therein lies the value of our study in which we also evaluated the effect in this group of participants in a prospective manner.

We observed that in our group of participants who followed dietary–hygienic measures, their 25(OH)D3 levels had decreased when measured a second time, which may result from the difficulty among the participants to closely adhere to the recommendations provided by the physician. Further, it is important to note that the second measurement of 25(OH)D3 was obtained in the winter season. These factors could explain the increased number of cases of SARS-CoV-2 infection and underscore the usefulness of supplementation in the population during this time of the year. Our regression model that adjusted the seasonal variation reinforces the importance of supplementation.

Since there is evidence of a greater risk of acquiring a SARS-CoV-2 infection and an increase in its severity among participants with 25(OH)D3 levels <20 ng/mL [8,9,25,26,27], we considered not randomizing-to-treatment the participants in these groups. They were referred to the primary care physician to initiate supplementation. Even with the treatment options and recommendations, some participants decided not to follow them. However, the supplemented group had a lower incidence of SARS-CoV-2 at four months of follow-up but not at six months. We believe that there was an important effect of 3-month supplementation, but it was not sustained through the six months follow-up, so future studies must be conducted where sustained supplementation is evaluated.

From a mechanistic point of view, vitamin D status could be an index of nitric oxide concentrations induced by solar UVA rays and these may act in concert to potentially prevent the COVID-19-dependent cytokine storm and induced inflammation [28,29,30]. Furthemore, vitamin D could protect against viral infection through the innate immune response. The induction of cathelicidin and defensins can block viral entry to the cell and suppress viral replication. Another mechanism is promoting autophagy of the virus, expressing autophagy marker LC3, downregulating the mTOR pathway, promoting Beclin 1 and PI3KC3, and stimulating the formation of autophagosomes to indirectly facilitate viral clearance. Therefore, vitamin D could have an important role in maintaining the balance between autophagy and apoptosis and thus maximize the antiviral response to infection [11,30].

Vitamin D is also a regulator of the adaptive immune response by inducing regulatory T cells that are critical to the induction of immune tolerance and play a role in preventing the cytokine storm associated with severe respiratory disease caused by viral infections [11].

Activation of the vitamin D receptor could play a modulatory role to the host response in the acute respiratory distress syndrome by decreasing cytokines, producing a shift toward amplified adaptive Th2 immune responses, regulating the renin-angiotensin-bradykinin system, modulating neutrophil activity, and maintaining the integrity of the pulmonary epithelial barrier, thus promoting epithelial repair, and decreasing the coagulability and prothrombotic tendency associated with SARS-CoV-2 infection [30,31,32].

One of our study’s limitations is the sample size in each group; however, using a formula for the proportion difference between independent groups of 26%, we achieved a power of 86%. Moreover, some participants did not complete their follow-up at 4- and 6-months because of SARS-CoV-2 vaccination. We decided to discontinue the study once patients were vaccinated since it could affect our primary outcome. The study was conducted in one center and mostly in the winter, so a multicenter study conducted during all seasons is required in the future, with a greater sample size to confirm the risks and benefits of vitamin D3 supplementation. Further studies must be conducted with single-dose supplementation during the hospital stay since previous studies had not demonstrated a difference [33]. We would like to further study a group with 25(OH)D3 levels ≥20 ng/mL without supplementation or diet/hygiene measures for comparisons. Furthermore, it is important to study if vitamin D3 is useful in preventing SARS-CoV-2 infection in subjects vaccinated against SARS-CoV-2. 

The implication of this study for the clinical practice, and academic and scientific community is to confirm that supplementation in subjects with 25(OH)D3 levels <20 ng/mL is useful for preventing SARS-CoV-2 infection. Furthermore, in individuals with blood levels of 25(OH)D3 levels ≥ 20 ng/mL that are at a high risk of exposure to SARS-CoV-2 or that are at increased risk of not following diet/hygiene measures, the supplementation with vitamin D3 could be helpful to reduce the risk of infection with SARS-CoV-2.

In conclusion vitamin D3 supplementation in participants with 25(OH)D3 levels between 20–100 ng/mL have a lower rate of SARS-CoV-2 infection compared with the use of dietary–hygienic measures at six months follow-up.

## Figures and Tables

**Figure 1 microorganisms-11-00282-f001:**
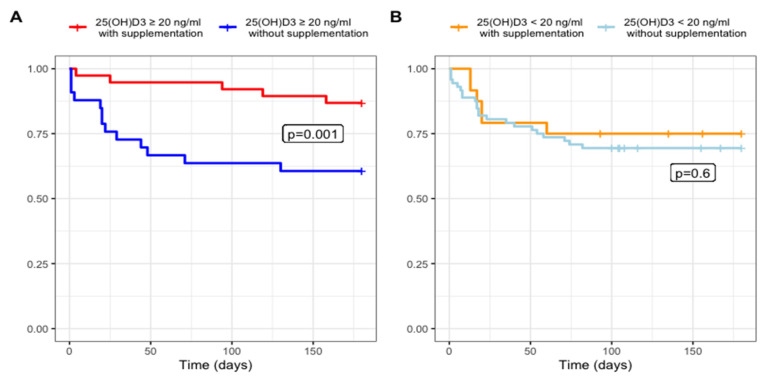
SARS-CoV-2-free survival at six-month follow-up. (**A**) 25(OH)D3 ≥20 ng/mL and (**B**) 25(OH)D3 ≤20 ng/mL.

**Table 1 microorganisms-11-00282-t001:** Demographics and Medical History.

Groups n = 198	G1	G2	*p*-Value	G3	G4	*p*-Value *
	n = 43	n = 42		n = 28	n = 85	
Female	26 (60)	19 (45)	>0.05	20 (71)	65 (76)	>0.05
Profession						
Physician (%)	15 (35)	16 (38)	>0.05	7 (25)	23 (27)	>0.05
Nurse (%)	19 (44)	15 (36)		13 (46)	39 (46)	
Dentist (%)	1 (2)	2 (5)		1 (4)	7 (8)	
Maintenance (%)	2 (5)	3 (7)		0	1 (1)	
Administrative (%)	5 (12)	1 (2)		4 (14)	9 (11)	
Nutritionist (%)	0	3 (7)		2 (7)	2 (2)	
Other (%)	1 (2)	2 (5)		1 (4)	4 (5)	
Medical History						
Tobacco(%)	1 (2)	4 (10)	>0.05	2 (7)	3 (4)	>0.05
BMI (SD)	26.4 (5.1)	26.3 (3.5)	>0.05	27.8 (2.1)	28.2 (4.5)	>0.05
Type 2 Diabetes (%)	5 (11)	1 (2)	>0.05	2 (7)	6 (7)	>0.05
Allergies (%)	7 (16)	7 (17)	>0.05	7 (25)	20 (24)	>0.05
Hypertension (%)	6 (14)	3 (7.5)	>0.05	5 (18)	11 (13)	>0.05
Asthma (%)	2 (5)	2 (5)	>0.05	0	0	NA

* Chi-square test was performed for group comparisons. A *p*-value ≤ 0.05 was considered statistically significant. Parentheses represent frequency or standard deviation. Abbreviation: SD: standard deviation; BMI: body mass index; G1: 25(OH)D3 ≥20 ng/mL with supplementation; G2: 25(OH)D3 ≥20 ng/mL without supplementation; G3: 25(OH)D3 <20 ng/mL with supplementation; G4 25(OH)D3 <20 ng/mL without supplementation.

**Table 2 microorganisms-11-00282-t002:** SARS-CoV-2 infection at 6-month follow-up.

n = 167	G1 (n = 38)	G2 (n = 33)	*p*-Value	G3 (n = 24)	G4 (n = 72)	*p*-Value
SARS-CoV-2	5 (13)	13 (39)	0.01	9 (37)	29 (40)	>0.05
Physician (%)	2 (15)	3 (23)	>0.05	2 (22)	5 (17)	>0.05
Nurse (%)	2 (11)	8 (61)	0.003	6 (67)	17 (59)	>0.05
Dentist (%)	0	0		0	3 (10)	
Administrative (%)	1 (20)	1 (100)	>0.05	0	3 (10)	
Nutritionist (%)	0	1 (33.)		1 (11)	1 (3)	>0.05
SARS-CoV-2 associated symptoms						
Dry cough (%)	0 (0)	0 (0)		2 (22)	5 (17)	>0.05
Fatigue (%)	2 (40)	1 (7.7)	>0.05	4 (44)	4 (14)	0.049
Sore throat (%)	1 (20)	0		1 (11)	5 (17)	>0.05
Nasal congestion (%)	1 (20)	1 (7.7)	>0.05	0	2 (7)	
Fever (%)	1 (20)	2 (15.4)	>0.05	1 (11)	2 (7)	>0.05
Myalgias (%)	4 (80)	2 (8)	0.009	4 (44)	4 (13.8)	0.049
Diarrhea (%)	1 (20)	1 (8)	>0.05	1 (11)	0	
Chest pain (%)	1 (20)	0		0	2 (7)	
Shaking chills (%)	1 (20)	0		1 (11)	0	
Nausea (%)	1 (20)	0		0	0	
Hyposmia (%)	0	1 (8)			0	
Hospitalizations (%)	2 (40)	1 (8)	>0.05	3 (33)	2 (7)	0.04
Intubations (%)	0 (0)	0 (0)		0	1 (3)	
Reservoir mask (%)	1 (20)	0 (0)		0	0	
Oxygen via nasal prongs (%)	0 (0)	0 (0)		0	1 (3)	
Pneumonia (%)	3 (60)	1 (7.7)	0.017	1 (11)	2 (7)	>0.05

A *p*-value ≤ 0.05 was considered statistically significant. Abbreviations: SD: standard deviation; G1: 25(OH)D3 ≥ 20 ng/mL with supplementation; G2: 25(OH)D3 D ≥20 ng/mL without supplementation; G3: 25(OH)D3 <20 ng/mL with supplementation; G4: 25(OH)D3 <20 ng/mL without supplementation. Parentheses represent frequency or standard deviation.

## Data Availability

Data are available upon reasonable request from the corresponding author.

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
