# Peer review of "Effect of Vitamin D3 Supplementation vs. Dietary–Hygienic Measures on SARS-CoV-2 Infection Rates in Hospital Workers with 25-Hydroxyvitamin D3 [25(OH)D3] Levels ≥20 ng/mL"

_microorganisms, 2023, doi:10.3390/microorganisms11020282_

Round 1
Reviewer 1 Report
First of all, I want to congratulate the authors for the study. In fact, the article reports an important study for the academic and scientific community. The study presents scientific and methodological criteria. After some changes, which are described later, it must be accepted for publication.
Line 20 - In the Abstract, it is important to clarify the type of study.
Line 70 – Why the option in a quasi-experimental study and not in a RCT?
Line 71 – the correct reference is number 25.
Are there no further limitations of the study carried out?
Are there implications of this study for clinical practice? And for the academic and scientific community? Can you specify?
Author Response
Thank you so much for your comments on improving our manuscript. Find highlighted in blue the corrections we made:
1. Line 20 - In the Abstract, it is important to clarify the type of study.
R= thank you, we added this information.
2. Line 70 – Why the option in a quasi-experimental study and not in a RCT?
R=We considered an quasi-experimental study because even though there is a comparative group and randomization, we could not blind treatment.
3. Line 71 – the correct reference is number 25.
R= thank you, we corrected as suggested.
4. Are there no further limitations of the study carried out?
We added this paragraph to the limitation section (page 8): Further studies must be conducted with single-dose supplementation during the hospital stay since previous studies had not demonstrated a difference [33]. We would like to further study a group with 25(OH)D3 levels >20 ng/mL without supplementation or diet/hygiene measures for comparisons. Also, it is important to study if vitamin D3 is useful in preventing SARS-COV2 infection in subjects vaccinated against SARS-COV-2.
5. Are there implications of this study for clinical practice? And for the academic and scientific community? Can you specify?
Thank you for your comments. We added this paragraph in the manuscript after limitations (page 8): The implication of this study for the clinical practice, academic and scientific community is to confirm that supplementation in subjects with 25(OH)D3 levels <20 ng/mL is useful for preventing SARS-COV2 infection. Also, in individuals with blood level of 25(OH)D3 levels >20 ng/mL that is at high risk of exposure to SARS-COV2 or that are at increased risk of not following diet/hygiene measures, the supplementation with vitamin D3 could be helpful to reduce infection of SARS-COV-2.
In conclusion vitamin D3 supplementation in participants with 25(OH)D3 levels between 20-100 ng/mL have a lower rate of SARS-COV-2 infection compared with the use of dietary-hygienic measures at six months follow-up.
Reviewer 2 Report
Nothing to disclose! I consider the paer deserves publication with its original form!
BR
Author Response
Thank you.